Comparative transcriptome analysis of differentially expressed genes related to the physiological changes of yellow-green leaf mutant of maize

Li Tingchun 1 2 litingchun2003@126.com
Yang Huaying 1
Lu Yan 3
Dong Qing 1
Liu Guihu 1
Chen Feng 2
Zhou Yingbing 1 ybzhou99@126.com
1 Tobacco Research Institute, Anhui Academy of Agricultural Sciences , Hefei , China
2 Department of Plant Sciences, University of Tennessee , Knoxville, TN , USA
3 Department of Biological Sciences, Western Michigan University , Kalamazoo, MI , USA
Maloof Julin
Electronic publication date: 2021 Feb 16
Publication date: 2021
Volume: 9
Electronic Location ID: e10567
Received 2020 Jun 16; Accepted 2020 Nov 23
Copyright: © 2021 Li et al.
Copyright year: 2021
Copyright holder: Li et al.
License: This is an open access article distributed under the terms of the Creative Commons Attribution License, which permits unrestricted use, distribution, reproduction and adaptation in any medium and for any purpose provided that it is properly attributed. For attribution, the original author(s), title, publication source (PeerJ) and either DOI or URL of the article must be cited.
License URL: https://creativecommons.org/licenses/by/4.0/

Keywords: Yellow-green leaf, Chlorophyll biosynthesis, Photosynthesis, Tricarboxylic acid cycle, Secondary metabolism, Transcriptome analysis

Funding: National Natural Science Foundation of China 31601240 Anhui Academy of Agricultural Sciences 2020YL058 This work was supported by the National Natural Science Foundation of China (31601240), the Scientific and Technological Innovation team of Anhui Academy of Agricultural Sciences (2020YL058) and the Fellowship of visiting scholar program of Anhui Academy of Agricultural Sciences (Tingchun Li, a visiting scholar at the University of Tennessee). The funders had no role in study design, data collection and analysis, decision to publish, or preparation of the manuscript.

==============================
Chlorophylls, green pigments in chloroplasts, are essential for photosynthesis. Reduction in chlorophyll content may result in retarded growth, dwarfism, and sterility. In this study, a yellow-green leaf mutant of maize, indicative of abnormity in chlorophyll content, was identified. The physiological parameters of this mutant were measured. Next, global gene expression of this mutant was determined using transcriptome analysis and compared to that of wild-type maize plants. The yellow-green leaf mutant of maize was found to contain lower contents of chlorophyll a, chlorophyll b and carotenoid compounds. It contained fewer active PSII centers and displayed lower values of original chlorophyll fluorescence parameters than the wild-type plants. The real-time fluorescence yield, the electron transport rate, and the net photosynthetic rate of the mutant plants showed reduction as well. In contrast, the maximum photochemical quantum yield of PSII of the mutant plants was similar to that of the wild-type plants. Comparative transcriptome analysis of the mutant plants and wild-type plants led to the identification of differentially expressed 1,122 genes, of which 536 genes were up-regulated and 586 genes down-regulated in the mutant. Five genes in the chlorophyll metabolism pathway, nine genes in the tricarboxylic acid cycle and seven genes related to the conversion of sucrose to starch displayed down-regulated expression. In contrast, genes encoding a photosystem II reaction center PsbP family protein and the PGR5-like protein 1A (PGRL1A) exhibited increased transcript abundance.

Introduction

Chlorophylls are essential pigments for photosynthesis, playing the main role in the conversion of light energy to stored chemical energy (Gitelson, Gritz & Merzlyak, 2003). Chlorophyll content directly determine photosynthetic potential and primary productivity of green plants (Gitelson, Gritz & Merzlyak, 2003; Curran, Dungan & Gholz, 1990; Filella et al., 1995). The formation of chlorophyll consists of four steps including synthesis of 5-aminolevulinic acid, formation of a pyrrole ring porphobilinogen, synthesis of protoporphyrin IX and insertion of Mg2+ to the protoporphyrin IX (Wu et al., 2007). The functional genes of the chlorophyll metabolism pathway have been identified.

Generally, the leaf color is green for its common content of chlorophyll. Nevertheless, a large number of leaf color mutants have been identified in many seed plant species, such as Arabidopsis, maize, soybean, barley, rice, and wheat (Wang et al., 2018). Among the leaf color mutants, a number of abnormal phenotypes have been identified, such as yellow, pale green, spots, and stripes. Due to reduced levels of Chlorophyll, retarded growth, dwarfism, and sterility were characterized in most color mutants. Recently, with the progressive characterization of various leaf color mutants, a significant number of genes have been isolated and verified to be responsible for the abnormal phenotype. For example, in carrot, a YEL locus was mapped in a linkage group with a total length of 33.2 cm, and the mutant had a yellow-leaf phenotype (Nothnagel & Straka, 2003). In maize, a semi-dominant oil yellow 1 (Oyl) mutant was identified to be deficient in the conversion of protoporphyrin IX to magnesium protoporphyrin IX (Sawers et al., 2006). The Oyl gene was demonstrated to encode I subunit of magnesium chelatase (ZmCHLI). In cabbage, the ygl-1 locus was located on chromosome C01. Mutation in the ygl-1gene exhibited a yellow-green leaf phenotype (Liu et al., 2016). In rice, mutation in the VIRESCENT YELLOW LEAF (VYL) gene, which encodes a subunit of chloroplast Clp (OsClpP6), resulted in temperature-insensitive and developmental stage-dependent virescent yellow leaf (vyl) phenotype (Dong et al., 2013). Fgl is located in the coding region of OsPORB, and its mutation resulted in the presence of the yellow/white leaf (Sakuraba et al., 2013). Mutation in the FdC2 gene, which encodes a ferredoxin-like protein with a C-terminal extension, caused the yellow-green leaf phenotype in rice (Li et al., 2015). Mutation in the rice YS83 (LOC_Os02g05890) gene resulted in the yellow-green-leaf phenotype as well (Ma et al., 2017). In addition, a number of chloroplast signal recognition particle (cpSRP) mutants were identified with Chlorophyll deficiency in Arabidopsis, rice and maize (Zhang et al., 2013; Guan et al., 2016; Wang & Grimm, 2016). Although these studies have provided many insights into the key genes controlling Chlorophyll deficiency, the analyses of physiological parameters of such mutants and their underlying molecular mechanisms have been generally lacking.

Chlorophyll-deficient mutants are important tools for studying the formation and development of photosynthetic pigments in plants. Their phenotypes could be used as crop trait markers in hybrid breeding (Zhong et al., 2015). In this work, a new yellow-green leaf mutant inbred line of maize was isolated. The Chlorophyll content, Chlorophyll fluorescence parameters, and photosynthesis characteristics were determined. Using comparative transcriptome analysis, differentially expressed genes were compared between the yellow-green leaf mutant and the normal green leaf inbred line. These results not only provide valuable genetic resources for further studies of Chlorophyll-deficient mutants in maize, but also contribute to our understanding of the relationship between physiological changes and gene expression changes. The latter may pave the way to further dissecting the molecular basis of morphological and physiological characteristics of the yellow-green leaf mutant.

Materials and Methods

Plant material

The maize inbred line C033 and LH102 were obtained from preserved breeds in rural areas of Anhui province which were stored in Tobacco Research Institute, Anhui province, People’s Republic of China. The yellow-green leaf mutant inbred line was isolated from an F2 segregating population of the recombination between inbred line C033 and LH102 in the farm of Tobacco Research Institute. After successive self-pollination of F2, F3 and F4 generations, a stable F5 generation was obtained. The yellow-green leaf mutant inbred line and a normal green leaf inbred line from the F5 generation were selected for downstream analyses. The inbred lines were cultivated with regular water and fertilizer management in the farm of Tobacco Research Institute in the year of 2017. After the third leaf was fully expanded, seedlings were selected for physiological parameter determination, RNA extraction, and gene expression analysis.

Measurement of chlorophyll content

Leaf samples (100 mg) were cut into small pieces, and soaked in 10 ml of 80% acetone (acetone: water = 4:1) at 4 °C in the dark for 24 h. The supernatant was collected after centrifugation at 6,000 rpm for 10 min. The absorbance was recorded at 663 and 645 nm on a UV-1800 spectrophotometer (Shimadzu Corporation, Kyoto, Japan). The concentrations of Chl a and Chl b were calculated using the method described by Arnon (1949). The values were calculated using three repeats.

Measurement of carotenoid compounds

To analyze the carotenoid compounds, 200 mg leaf samples were grinded into powder with 2 ml absolute alcohol containing 1% butylated hydroxytoluene. After water bath for 5 min at 85 °C, 40 ul 80% KOH and 1 ml N-hexane were added into the extraction buffer followed by water bath and vortex. The supernatant were eventually collected and dried with nitrogen, then dissolved in 500 ul acetonitrile solution containing 1% butylated hydroxytoluene, 25% methanol, and 5% dichloromethane for following analysis.

The Ultimate 3000 UHPLC system (Thermo Fisher Scientific, Waltham, MA, USA) was employed to quantitatively and qualitatively determine the components. Carotenoids were resolved and analyzed on a reverse phase YMC carotenoid column (250 * 4.6 mm, 5 um; YMC, Kyoto, Japan) set at a temperature of 40 °C with the flow rate of 1 ml·min−1. The solvent system consisted of solvent A with methanol: methyl tert-butyl ether: water (81:15:4, by vol) and solvent B with methanol: methyl tert-butyl ether (6.5:93.5, by vol). The gradient program was set as follows: 2 min hold on 100% solvent A, followed a 1 min linear gradient to 32.5% solvent A and 67.5% solvent B, then 2 min hold on 100% solvent B, and 2 min hold on 100% solvent A lastly. Carotenoid compounds were detected at 450 nm. The determination was repeated three times.

Measurement of the total starch, total sugar and enzymes activities

The contents of the total starch, the total soluble sugar and the total reducing sugar were measured using the methods described by Mccleary et al. (1994), Irigoyen, Emerich & Sanchez-Diaz (1992) and Miller (1959). The enzymes activities of SS (Sucrose synthase) and SPS (Sucrose phosphate synthase) were assayed according to Echeverria and Humphreys’ method (Echeverria & Humphreys, 1985). The enzymes activities of SSS, GBSS and AGP were determined using the methods described by Wang et al. (2007) and Nakamura, Yuki & Park (1989). All the measurement of physiological parameters was repeated three times.

Measurements of chlorophyll fluorescence parameters

By using the PAM-2500 chlorophyll fluorometer (Walz, Effeltrich, German), chlorophyll fluorescence parameters were determined for the leaves from the yellow-green leaf mutant inbred line and the normal green leaf inbred line. The procedure was described as follows. After the leaf was adapted in the dark with the dark adapting clip for 20 min, the slow kinetics of chlorophyll a fluorescence induction was triggered with a continuous mode to measure dark- and light-adapted parameters. The leaf was initially subjected to a measuring light of 95 μmol photons·m−2·s−1. After Fo was recorded, a saturating pulse of 2,000 μmol photons·m−2·s−1 was automatically turned on, and Fm was measured accordingly. At that time, an actinic light of 145 μmol photons·m−2·s−1 was activated to simulate normal irradiance conditions. F, Fm′ and Fo′ were subsequently measured with saturating pulses every 20 s. The parameters ФPSII, ФNPQ, ФNO, NPQ, qN, qP, qL and ETR were derived from the final measurements obtained after a 10 min light adaptation.

The maximum photochemical quantum yield of PSII (Fv/Fm) was calculated according to Stefanov & Terashima (2008): Fv/Fm = (Fm − Fo)/Fm.

The effective photochemical quantum yield of PSII (ФPSII), the quantum yield of non-regulated heat dissipation and fluorescence emission (ФNO), and quantum yield of light-induced non-photochemical quenching (ФNPQ) were calculated as described by Kramer et al. (2004): ФPSII = (F′m − F)/F′m, ΦNO = 1/(NPQ + 1 + qL (Fm/Fo − 1)), ФNPQ = 1 − ФPSII − (1/(NPQ + 1 + qL (Fm/Fo − 1))).

The coefficient of photochemical quenching (qL) was calculated as described by Kramer et al. (2004): qL = qP × F′o/F.

The coefficients of photochemical quenching (qP), non-photochemical quenching (qN and NPQ) were calculated as described by Stefanov & Terashima (2008): qP = (F′m − F)/(F′m − Fo), qN = 1 − (Fm − F′m)/(Fm − F′o), NPQ = (Fm − F′m)/F′m.

The relative apparent photosynthetic electron transport rate (ETR) was calculated using the equation: ETR = ФPSII × PAR × 0.5 × 0.84.

Measurement of photosynthesis parameters

The net photosynthetic rate (Pn) was measured using a Li-6400XT portable photosynthesis system (Li-Cor Inc., Lincoln, NE, USA) equipped with a 6400-02B chamber and a red-blue LED light source with intensities up to 2,000 μmol photons·m−2·s−1 over an area of 6 cm2. The flow rate was adjusted to 500 μmol·s−1 with the absolute CO2 concentration of 380 μmol·mol-1 at 26 °C inside the chamber. The light response curve of Pn was determined at nine photosynthetically active radiation (PAR) levels (0, 50, 100, 200, 400, 800, 1,200, 1,600 and 2,000 μmol photons·m−2·s−1). Three biological repeats were created.

cDNA library construction and sequencing

By using Illumina TruSeq™ RNA Sample Preparation Kit (Illumina, San Diego, CA, USA), the cDNA library was constructed. After the quality detection, the Illumina sequencing was carried out at Beijing Novogene Biological Information Technology Co. Ltd. (Beijing, China) (http://www.novogene.cn/). The index-coded samples were clustered following the manufacturer’s instructions using TruSeq PE Cluster Kit v3-cBot-HS (Illumina, San Diego, CA, USA). Then, the library was sequenced to generate 200 bp paired-end reads on an Illumina Hiseq 2500 platform. The raw data was accessible at the Sequence Read Archive Database of NCBI (https://www.ncbi.nlm.nih.gov/) with the accession number PRJNA629016.

Data filtering and assembly

The clean reads were obtained after remove of duplicated sequences, ploy-N, adaptor sequences and low-quality reads. Then, they were aligned to the the maize B73 reference genome (AGPv4) using TopHat 2 as previously described (Schnable et al., 2009; Kim et al., 2013). The resulting read counts were normalized by per kilobase million mapped reads (RPKM) to measure the gene expression level (Mortazavi et al., 2008). Maize reference genome sequence data were downloaded at https://www.maizegdb.org/genome/assembly/Zm-B73-REFERENCE-GRAMENE-4.0.

Identification of differential expressed genes

To identify differentially expressed gene (DEGs) in the comparison settings, the raw counts were imported into the edgeR and adjusted with one scaling normalized factor (Zhang et al., 2014). Then the DEGseq R package (1.12.0; TNLIST, Beijing, China) was employed to screen out the DEGs with P-value < 0.05 (Benjamini & Hochberg, 1995). The sequences of all DEGs were listed in Table S6.

Gene functional annotation and metabolic pathway analysis

The DEGs were annotated via alignment against and comparison with Pfam (http://pfam.xfam.org/), NCBI non-redundant (Nr) protein database (https://www.ncbi.nlm.nih.gov/refseq/), and SwissProt protein database (https://web.expasy.org/docs/swiss-prot_guideline.html). The GO terms including molecular function, biological process, and cellular component ontology were analyzed using Blast2GO program. Pathway assignments were conducted according to the Kyoto Encyclopedia of Genes and Genomes Pathway database (KEGG, http://www.genome.jp/kegg). To map the target genes to metabolic pathways, all sequences of DEGs were uploaded to the Mercator v.3.6 (https://www.plabipd.de/portal/web/guest/mercator-sequence-annotation) to generate root map file, then it was imported to the Mapman software (V3.6.0 RC1) to obtain the map based on the transcriptome data (Kakumanu et al., 2012).

Verification of unigenes and gene expression profiling using RT-qPCR

Quantitative Real-Time PCR was performed to quantify nine DEGS to evaluate the validity of transcriptome data. The candidate genes and their primers are listed in Table S1. The RNAprep Pure Plant Kit (Tiangen, Beijing, China) was used to obtain the total RNA. The PCR system contained 2 μl primers, 2 μl of cDNA, 8.5 μl of ddH2O, and 12.5 μl of SYBR® Premix Ex TaqTM II. PCR amplifications follow the procedure, 95 °C for 30 s, 95 °C for 5 s, 60 °C for 30 s, 40 cycles. Quantification was calculated with the 2−ΔΔCt method (Livak & Schmittgen, 2001).

Results

Identification of a maize yellow-green leaf mutant and measurement of its pigment contents

A yellow-green leaf mutant inbred line was firstly isolated from an F2 segregating population of the cross recombination inbred line C033 and inbred line LH102, both of which have a green leaf phenotype. After successive self-pollination of F2, F3 and F4 generations, a stable F5 generation was obtained. In this study, a stable yellow-green leaf mutant inbred line and a normal green leaf inbred line from the F5 generation were selected for downstream characterizations. The yellow-green leaf mutant had a yellow color in the entire above-ground portion of the plant (Fig. 1A).

Figure 1 Phenotypic characteristics change of yellow-green leaf mutant plants.

(A) Indicated two yellow-green leaf mutant plants and a normal green leaf plant at the same age. Maize plants in this image were at the five-leaf stage. (B) Showed the contents of chlorophyll a and chlorophyll b in the normal green leaf inbred line and the yellow-green leaf mutant. (C) showed the contents of eight carotenoid compounds including neoxanthin, violaxanthin, capsanthin, zeaxanthin, β-cryptoxanthin, ɑ-carotene, β-carotene and lutein. Small letters a and b above the columns indicate differences between the yellow-green leaf mutant and the normal green leaf inbred line at P < 0.05, according to least significant difference (LSD) tests. FW is the abbreviation of the fresh weight.

Leaf color could indicate the amount and proportion of chlorophyll in leaves. Deficiency of chlorophyll leads to the leaf color change from green to yellow. In this study, in contrast with the normal green leaf inbred line, the content of chlorophyll a and chlorophyll b in the yellow-green leaf mutant was reduced by 35.22% and 34.48%, respectively, which may directly result in the presence of yellow-green color in the mutant plants (Fig. 1B). Otherwise, the contents of seven kinds of carotenoid compounds including neoxanthin, violaxanthin, capsanthin, zeaxanthin, α-carotene, β-carotene and lutein were significantly decreased in the yellow-green leaf mutant (Fig. 1C).

Measurement of chlorophyll fluorescence parameters of the yellow-green leaf mutant

For further analysis, chlorophyll fluorescence parameters were determined to evaluate the changes of light absorption and energy transfer in the light-harvesting complexes. Fo indicates the minimum fluorescence yield after dark-adaptation with all PSII centers open. Fm represents the maximum fluorescence yield after dark-adaptation with all PSII centers closed. Both Fo and Fm were decreased in the yellow-green leaf mutant, suggesting that the yellow-green leaf mutant has fewer active PSII centers than the normal green leaf inbred line (Fig. 2A). However, the yellow-green leaf mutant and the normal green leaf inbred line had the similar value of the maximum photochemical quantum yield of PSII (Fv/Fm), suggesting that light absorption and energy transfer of the light-harvesting complexes is still efficient in the yellow-green leaf mutant plants.

Figure 2 Chlorophyll fluorescence and photosynthesis parameters in the yellow-green leaf mutant inbred line and the normal green leaf inbred line.

(A) Indicated the original chlorophyll fluorescence parameters. (B) Showed real-time fluorescence yield Ft in the normal green leaf inbred line and the yellow-green leaf mutant. The point pointed by the arrows indicated the turned on of the actinic light. (C) Indicated additional chlorophyll fluorescence parameters of the normal green leaf inbred line and the yellow-green leaf mutant. Small letters a and b indicate differences between the yellow-green leaf mutant and the normal green leaf inbred line at P < 0.05, according to least significant difference (LSD) tests. (D) Showed light response curves of net photosynthesis in the yellow-green leaf mutant and the normal green leaf inbred line. Pn is the net photosynthesis rate. The light response curves were measured at nine PAR levels (0, 50, 100, 200, 400, 800, 1,200, 1,600 and 2,000 μmolphotons·m−2·s−1). The dark respiration rate is defined as the Pn value when the light response curve intersects the Y-axis. The measurement was performed for three times. The error bars represented the standard errors.

Ft is the real-time fluorescence yield recorded during the slow kinetics induction with the continuous monitoring mode. The changes of Ft reflect the light-adaption status of the PSII centers. Although the Ft kinetics curve in the yellow-green leaf mutant had the same light-adaption pattern as that in the normal green leaf inbred line, the Ft values were much lower in the yellow-green leaf mutant (Fig. 2B). Accordingly, the values of the minimal and maximal fluorescence yield in the light-adapted state (Fo′ and Fm′) were significantly lower in the yellow-green leaf mutant than in the normal green leaf inbred line (Fig. 2C). Photochemical quenching parameters (ФPSII, qP, and qL), non-photochemical quenching parameters (ФNO, ФNPQ, NPQ, and qN), and the PSII electron transport rate (ETR) were also evaluated. The values of ФPSII, qP, qL, ФNO, ФNPQ, NPQ, and qN were similar between the yellow-green leaf mutant and the normal green leaf inbred line (Fig. 2C). But the value of ETR was significantly lowered in the yellow-green leaf mutant.

Net photosynthesis in response to light intensities of the yellow-green leaf mutant

Net photosynthesis (Pn) in response to different light intensities was also determined (Fig. 2D). Pn was lower in the yellow-green leaf mutant than in the normal green leaf inbred line when the light intensity was 1,200 μmol photons·m−2·s−1or higher. But the rate of dark respiration was higher in the yellow-green leaf mutant.

Comparative transcriptome analysis of the yellow-green leaf mutant and the normal green leaf inbred line: overall changes

To explain the physiological changes in the yellow-green leaf mutant, comparative transcriptome analysis was performed to identify differentially expressed genes in chlorophyll biosynthesis, light-harvesting antenna complex formation, photosynthesis, and other metabolism pathways. Three independently repeated sequencing of cDNA was carried out by using the HiSeq 2500 platform. After data processing, the clean reads were obtained. Pearson correlation was calculated for the three independent experiments. The data from NGL_1, NGL_3, YGL_2, and YGL_3 were collected for the following analysis for the higher correlation rate between the same color leaves (Fig. S1). Further studies were performed to identify differentially expressed genes, using DGE methods. A total of 1122 genes were found to be differentially expressed in the normal green leaf inbred line and the yellow-green leaf mutant: 536 genes were up-regulated, and 586 genes were down-regulated in the yellow-green leaf mutant (Fig. 3A; Table S2). Most of these genes were enriched in biological processes and molecular functions (Table S3). Based on the KEGG pathway analysis and pathway assignment of MapMan, of all these differentially expressed genes, 1,092 genes were mapped to 33 metabolic pathways, including photosynthesis (14 genes), lipid metabolism (40 genes), secondary metabolism (52), stress response (55), and so on (Fig. 3B; Tables S3 and S4). The remaining 328 genes were not assigned.

Figure 3 Differentially expressed genes and its enriched metabolic pathways between the yellow-green leaf mutant and the normal green leaf inbred line.

(A) Indicated differentially expressed genes between the yellow-green leaf mutant and the normal green leaf inbred line. Differentially expressed genes were selected by q-value < 0.005 & |log2 (fold change)| > 1. The X axis indicates gene expression changes in different samples, and the Y axis indicates the significant degree of gene expression changes. Scattered points represent each gene, the red dots represent differentially up-regulated genes, the green dots represent differentially down-regulated genes, and the blue dots represent no significant difference gene. YGL, yellow-green leaf mutant; NGL, normal green leaf inbred line; −log10 (padj), the corrected p-value (padj < 0.05). (B) Showed the pie chart of enriched metabolic pathways of genes differentially expressed in the yellow-green leaf mutant and the normal green leaf inbred line. The pie chart was generated by submission of the differentially expressed genes to the online Mercator sequence annotation tool (http://www.plabipd.de/portal/mercator-sequence-annotation).

Differentially expressed genes in chlorophyll biosynthetic pathways

Chlorophylls are synthesized from the precursor protophorphyrin IX. As shown in Fig. 4, three chlorophyll metabolic genes (magnesium-chelatase subunit chlD (Zm00001d013013), protochlorophyllide reductase A (Zm00001d001820)), and chlorophyllide a oxygenase (Zm00001d004531) showed decreased expression levels in the yellow-green leaf mutant. Similarly, two protophorphyrin biosynthetic genes (coproporphyrinogen III oxidase (Zm00001d026277) and uroporphyrinogen decarboxylase 1 (Zm00001d044321)) displayed decreased expression in the yellow-green leaf mutant. However, genes encoding chlorophyllide a oxygenase (Zm00001d002358) and protoporphyrinogen oxidase (Zm00001d 008203) showed increased transcript abundance in the yellow-green leaf mutant.

Figure 4 Differentially expressed genes involved in chlorophyll metabolism between the yellow-green leaf mutant and the normal green leaf inbred line.

Differentially expressed genes in photosynthetic reactions

As shown in Fig. 5, 22 genes in photosynthesis were differentially expressed in the yellow-green leaf mutant and the normal green leaf inbred line. Of these genes, two are related to photosynthetic light reactions, 11 genes participate in photorespiration, and nine are involved in the Calvin cycle. Between the two genes in photosynthetic light reactions, one encodes a Mog1/PsbP/DUF1795-like photosystem II reaction center PsbP family protein (Zm00001d041824) and the other encodes PGRL1A (PGR5-like protein 1A, Zm00001d034904). These two genes exhibited increased transcript abundance in the yellow-green leaf mutant. However, genes encoding two ribulose bisphosphate carboxylase/oxygenase (RuBisCO) large chain precursors (Zm00001d00279 and GRMZM5G815453), which are involved in the Calvin cycle and photorespiration, showed decreased transcript abundance in the yellow-green leaf mutant. Genes encoding two RuBisCO large subunit-binding protein subunit alpha (Zm00001d031503 and Zm00001d051252) displayed decreased transcript abundance in the yellow-green leaf mutant as well. Genes encoding chloroplast chaperonin 60 subunit beta (Zm00001d035937), RuBisCO methyltransferase family protein (Zm00001d020437), TCP/cpn60 chaperonin family protein (Zm00001d045544), and RuBisCO large subunit-binding protein subunit alpha (Zm00001d00399) showed increased transcript abundance in the yellow-green leaf mutant. The expression of phosphoglycolate phosphatase (Zm00001d034887) and glycerate dehydrogenase (Zm00001d014919) was enhanced in the yellow-green leaf mutant, whereas the expression of glycine dehydrogenase (Zm00001d023437) and aldolase superfamily protein (Zm00001d040084) was decreased.

Figure 5 Differentially expressed genes involved in photosynthesis light reactions and carbon reactions between the yellow-green leaf mutant and the normal green leaf inbred line.

Differentially expressed genes in the tricarboxylic acid cycle

The tricarboxylic acid cycle (TCA cycle) is responsible for the production of most of the ATP yield. As shown in Fig. 6, a total of nine genes in the TCA cycle were down-regulated in the yellow-green leaf mutant. Among these genes, pyruvate phosphate dikinase (Zm00001d010321) catalyzes the conversion of pyruvate to phosphoenolpyruvate (PEP), aconitatehydratase (Zm00001d015497) catalyzes the stereo-specific isomerization of citrate to isocitrate, and isocitrate dehydrogense (Zm00001d025690) catalyzes the conversion of isocitrate to alpha-ketogutarate and CO2. The other six genes are involved in oxidative phosphorylation: NADH-ubiquinone oxidoreductase 20 kDa subunit (Zm00001d043619), NADH dehydrogenase (Zm00001d016864), ubiquinol-cytochrome c reductase iron-sulfur subunit (Zm00001d016619), cytochrome c (Zm00001d042600), member of uncoupling protein PUMP2 family (Zm00001d048583), and cytochrome c oxidase (Zm00001d051055).

Figure 6 Differentially expressed genes in the tricarboxylic acid cycle between the yellow-green leaf mutant and the normal green leaf inbred line.

Differentially expressed genes in the sucrose-to-starch pathway

In Fig. 7, nine genes in the sucrose-to-starch pathway were found to be differentially expressed between the yellow-green leaf mutant and the normal green leaf inbred line. Among these genes, two were up-regulated and seven were down-regulated in the yellow-green leaf mutant. Among the two up-regulated genes, one encodes aglycosyl hydrolase family 32 protein (Zm00001d025943), which may function as the invertase that split sucrose into glucose and fructose. The other gene is annotated as fructokinase-like protein (Zm00001d033181), which may catalyze the conversion of fructose to fructose-6-phosphate. Among the seven down-regulated genes, three are annotated as granule-bound starch synthase 1b (Zm00001d027242, Zm00001d029360, and Zm00001d019479), two encode soluble starch synthase (Zm00001d0002256 and Zm00001d0045261) and one encodes 1, 4-alpha-glucan branching enzyme IIB (Zm00001d003817). These six genes participate in starch synthesis. The other down-regulated gene is annotated as alpha-1, 4 glucanphosphrylase L isozyme (Zm00001d034074), which catalyzes the conversion of starch to glucose-1-phosphate.

Figure 7 Differentially expressed genes involved in sucrose to starch conversion between the yellow-green leaf mutant and the normal green leaf inbred line.

Validation of unigenes and gene expression profiling

Nine candidate genes were selected to test the validity of transcriptome date using RT-qPCR. The results showed that the expression patterns of nine genes determined using RT-qPCR were consistent with the transcriptome data, indicating the transcriptome data were very reliable (Additional file: Fig. S4).

Discussion

In maize, a yellow-green leaf mutant SN62 has been identified (Zhong et al., 2015). Photosynthetic characteristics of SN62 revealed that its chlorophyll content, the quantum efficiency of PSII and maximal quantum yield of PSII photochemistry were significantly lower than those of a medium-green leaf inbred line SN12 (Zhong et al., 2015). In this study, another yellow-green leaf mutant was identified. The values of photosynthetic parameters in this newly identified yellow-green leaf mutant were comparable to those in the previously identified yellow-green leaf mutant (SN62). The chlorophyll content and the values of chlorophyll fluorescence parameters (Ft, Fo, Fm, Fo′, Fm′ and ETR) in the yellow-green leaf mutant were significantly lowered than those in the normal green leaf inbred line. These data indicate that the yellow-green leaf mutant has fewer opened PSII reaction centers than the normal green leaf inbred line. However, the yellow-green leaf mutant and the normal green leaf inbred line had the similar value of the maximum photochemical quantum yield of PSII (Fv/Fm). This suggests that light absorption and energy transfer of the light-harvesting complexes is still efficient in the yellow-green leaf mutant plants. In addition, there were no obvious differences in ΦPSII, ΦNO, ΦNPQ, NPQ, qN, qP, and qL between the yellow-green leaf mutant and the normal green leaf inbred line. Furthermore, Pn was only lower in the yellow-green leaf mutant than in the normal green leaf inbred line when the light intensity was at 1,200 μmol photons·m−2·s−1 or higher.

In seed plants, most of the genes responsible for the chlorophyll biosynthesis pathway have been identified (Tripathy & Pattanayak, 2012). Magnesium-protoporphyrin chelatase catalyzes the first step in chlorophyll synthesis. This enzyme contains three subunits (ChlH, ChlD and ChlI) and catalyzes the insertion of Mg2+ into protoporphyrin IX. Mutation in ChlD resulted in a chlorina (yellowish-green) phenotype in rice (Zhang et al., 2006). Protochlorophyllide reductase catalyzes the conversion of pchlide to chloro-phyllide. Chlorophyllide a oxygenase (CAO) is responsible for chlorophyll b biosynthesis (Reinbothe et al., 2006). Overexpression of CAO was found to enlarge the antenna size of photosystem II in Arabidopsis (Tanaka et al., 2001). Mutation in the barley CAO (fch2) gene leads to chlorophyll b deficiency, which may affect electron transfer in photosystem II (Mueller et al., 2012). Coproporphyrinogen III oxidase is a key enzyme in the biosynthetic pathway of chlorophyll and heme. The deficiency in coproporphyrinogen III oxidase caused lesion formation in Arabidopsis (Ishikawa et al., 2001). Uroporphyrinogen decarboxylaseis is responsible for the decarboxylation of four acetate groups of uroporphyrinogen III to yield coproporphyrinogen III, resulting in heme and chlorophyll biosynthesis (Fan et al., 2007). In this study, these genes were found to have decreased transcript abundance in the yellow-green leaf mutant, which may directly cause chlorophyll deficiency and reduced formation of light-harvesting antenna complexes.

Photosynthesis begins with the light reactions. In this work, the gene encoding Mog1/PsbP/DUF1795-like photosystem II reaction center PsbP family protein and the gene annotated as PGRL1A were found to have increased transcript abundance. PsbP is necessary for the retention of Ca2+ and Cl−1, the assembly of PSII complex, and the maintenance of normal thylakoid architecture in PSII (Cao et al., 2015). PGRL1A is associated with PSI and it interacts with PGRL1 (DalCorso et al., 2008). The PGRL1-PGR5 complex was found to facilitate cyclic electron flow (DalCorso et al., 2008). In this work, the expression of both genes was up-regulated in the yellow-green leaf mutant, which may make the fewer opened PSII reaction centers work efficiently in the yellow-green leaf mutant.

Tricarboxylic acid cycle plays a central role in generating ATP and providing reducing agent NADH and precursors for a number of amino acids in both heterotrophic and photosynthetic tissues (Daloso et al., 2015). In this work, totally nine genes were identified with down-regulated expression in the yellow-green leaf mutant, including pyruvatephosphate dikinase, aconitatehydratase, isocitrate dehydrogenase, NADH dehydrogenase, and NADH-ubiquinone oxidoreductase 20 kDa subunit. Pyruvate phosphate dikinase is the key enzyme in cellular energy metabolism; it catalyzes the ATP- and phosphate (Pi)-dependent conversion of pyruvate to phosphoenol pyruvate in C4 plants (Ciupka & Gohlke, 2017). Aconitate hydratase catalyzes the conversion of citrate to cis-aconitate (Lichardusova et al., 2017). Isocitrate dehydrogenase catalyzes oxidative decarboxylation of isocitrate (Mhamdi & Noctor, 2015). The other six genes are involved in the mitochondrial electron transport chain and ATP synthesis, which requires the participation of large protein complex I (NADH-ubiquinone oxidoreductase), II (NADH dehydrogenase), III (ubiquinol-cytochrome c reductase) and IV (cytochrome c oxidase) (Møller, 2002; Dudkina et al., 2005). The decreased expression of these genes may have negative effects on ATP generation in the mitochondria of the yellow-green leaf mutant.

Furthermore, seven genes involved in the conversion of sucrose to starch were found to have decreased transcript abundance in the yellow-green leaf mutant. Of these genes, granule-bound starch synthase 1b is responsible for amylose synthesis (Suzuki et al., 2015). Soluble starch synthase is a key enzyme in the biosynthesis of amylopectin (Wang et al., 2017). Moreover, the analysis of enzymes activities in starch biosynthesis pathway further demonstrated the results. The enzymes activities of SS, SSS, SPS and GBSS were significantly lower in yellow-green leaf if compared with that in normal green leaf (Additional file: Fig. S2). But there were no obvious difference in the contents of starch and total reducing sugar, though the content of the water-soluble total sugar was high in normal green leaf (Additional file: Fig. S3).

Conclusions

In summary, the yellow-green mutant leaf was identified with obviously lowered chlorophyll content. The phenotype changes directly caused the decrease of light absorption and energy transfer, photosynthesis, and starch synthesis. Comparative transcriptome analysis identified the differentially expressed gene between the yellow-green leaf and normal green leaf. Further analysis revealed that the changes of genes expression were consistent with the variation of physiological data. The downregulated expression of genes in chlorophyll biosynthesis pathway resulted in chlorophyll deficiency in yellow-green mutant leaf. Then, it negatively affected the expression of genes in photosynthesis, TCA cycle, starch biosynthesis, and so on. The enzymes activities, the net photosynthesis rate, and water-soluble sugar were eventually decreased in yellow-green mutant leaf. These findings provide potential explanations for observed morphological and physiological changes in the yellow-green leaf mutant. Further investigations are needed to unravel the molecular basis of the morphological, physiological and transcriptional changes in the yellow-green leaf mutant plants.

Supplemental Information

Supplemental Information 1 Primers for qRT-PCR analysis.

Click here for additional data file.

Supplemental Information 2 The list of differentially expressed genes between yellow-green leaf and normal green leaf.

Click here for additional data file.

Supplemental Information 3 GO terms analysis of differentially expressed genes.

Click here for additional data file.

Supplemental Information 4 KEGG pathway analysis of differentially expressed genes.

Click here for additional data file.

Supplemental Information 5 Pathway assignment of differentially expressed genes using MapMan software.

Click here for additional data file.

Supplemental Information 6 The sequences of all differentially expressed genes.

Click here for additional data file.

Supplemental Information 7 The Pearson correlation between individual RNA samples from the yellow-green leaf mutant and the normal green leaf inbred line.

The R2 value of the Pearson correlation between each pair of samples is presented in the center of each square. NGL_1, NGL_2 and NGL_3 are the three repetitions of the normal green leaf inbred line. YGL_1, YGL_2 and YGL_3 are the three repetitions of yellow-green leaf mutant.

Click here for additional data file.

Supplemental Information 8 The activities of enzymes SS, SSS, SPS, GBSS and AGP involved in starch biosynthesis pathway.

Lowercase letters a and b above the columns indicate differences between the yellow-green leaf mutant and the normal green leaf inbred line at P<0.05, according to least significant difference (LSD) tests.

Click here for additional data file.

Supplemental Information 9 The contents of starch, total reducing sugar and the water-soluble total sugar.

Click here for additional data file.

Supplemental Information 10 Validation of unigenes and DGE genes expression profiling.

Click here for additional data file.

Supplemental Information 11 Raw data of physiological indexes and relative expression level of genes.

Click here for additional data file.

Additional Information and Declarations

Competing Interests

Author Contributions

Data Availability

The authors declare that they have no competing interests.

Tingchun Li conceived and designed the experiments, performed the experiments, analyzed the data, prepared figures and/or tables, authored or reviewed drafts of the paper, and approved the final draft.

Huaying Yang conceived and designed the experiments, performed the experiments, prepared figures and/or tables, and approved the final draft.

Yan Lu conceived and designed the experiments, analyzed the data, prepared figures and/or tables, authored or reviewed drafts of the paper, and approved the final draft.

Qing Dong performed the experiments, analyzed the data, prepared figures and/or tables, and approved the final draft.

Guihu Liu performed the experiments, prepared figures and/or tables, and approved the final draft.

Feng Chen conceived and designed the experiments, analyzed the data, authored or reviewed drafts of the paper, and approved the final draft.

Yingbing Zhou conceived and designed the experiments, analyzed the data, authored or reviewed drafts of the paper, and approved the final draft.

The following information was supplied regarding data availability:

Raw data is accessible at the Sequence Read Archive: PRJNA629016.

Other data are available in the Supplemental Files.

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
