# Peer review of "Comparative transcriptome analysis of differentially expressed genes related to the physiological changes of yellow-green leaf mutant of maize"

_PeerJ, doi:10.7717/peerj.10567_

## Round 0.1 · original submission · Major Revisions

I carefully read the submission titled ‘Comparative transcriptome analysis of differentially expressed genes related to the physiological changes of yellow-green leaf mutant of maize’. The manuscript would be of general interest; however, the following points from Reviewers must be addressed. The specific reviewers' comments are attached.

Reviewer 1 ·

Basic reporting

no comment

Experimental design

no comment

Validity of the findings

no comment

Additional comments

The manuscript “Comparative transcriptome analysis of differentially expressed genes related to the physiological changes of yellow-green leaf mutant of maize” identified a yellow-green leaf mutant of maize and analyzed the differentially expressed genes related to the physiological changes by transcriptome. The methodologies used are appropriated and results are well discussed. However, there are some points that could be improved:
1. In line 333, Fig.S3 should be revised as Fig.S2, and in line 335, Fig.S4 should be revised as Fig.S3
2. Supplementary Table 2, 3 and 4 didn’t mention in the MS.
3. Did the authors upload the transcriptome raw data? The accession number of the transcriptome raw data should be added in the MS.
4. Gene name should be italic, for example, in line 228, “Zm00001d013013” should be “Zm00001d013013”.
5. Formats of references should be in accordance with the requirements specified in peerj.

·

Basic reporting

This manuscript described the difference in physiological and transcriptomic characterization between the yellow-green leaf mutant and the normal green leaf inbred line of maize. Physiological experiments on photosynthesis parameters revealed that the yellow-green leaf mutant of maize contained fewer active PSII centers with lowered net photosynthesis rate, but have the similar efficiency of the light absorption and energy transfer of the light-harvest complex with that of the wild-type plants. And, Comparative transcriptomic analysis of the mutant plants and wild-type plants led to the identification a lot of differentially expressed genes response to their physiological change. These results not only provided valuable genetic resources for further studies of chlorophyll-deficient mutants in maize, but also contribute to our understanding of the molecular basis of morphological and physiological characteristics of the yellow-green leaf mutant.
In general, this is a well written and well organized manuscript, but some corrections and observations indicated as follows must be attended.

Experimental design

There are many physiological and biochemical indexes were determined, including chlorophyll content, chlorophyll a fluorescence parameters, photosynthesis parameters, and so on. The author should clearly state if there are biological repeats were created for each experiment. It is the necessarily case the scientific experiments.

Validity of the findings

All underlying data have been provided. They are robust, statistically sound and controlled. Conclusions are well stated.

Additional comments

Authors should prepare the corrected manuscript following very carefully the Instructions to Authors Guide. There are quite some evident grammatical errors or typos, which need to be intensively examined and corrected.
Line 12, it seems that the author provided the wrong email address for the corresponding author.
Line 61, the “YS83 (LOC_Os02g05890)” should be written in italic. The same mistake were found in the following line 62, the cpSRP should be in italic too.
Line 100, line 98-105, there should be a space between the number and the unit. For example, 250*4.6mm, 5um, 40℃, the author should be more careful to avoid similar mistakes
Line 172-173, the authors should briefly state the details of the method to map the target genes to metabolic pathways using the MapMan software.
Line 210 the unit “photonsm-2s-1” was wrote in the wrong way.
Line 182 there is should be an indentation at the beginning of the second graph.
Line 208, 222, 223, and 236, there should be a space between Fig. and numbers.
Line 223-225, the numbers stated in the text are inaccurate, the authors should clearly state and make the result more easier to be understood.
Line 253-262, there are many typos and some inaccuracies about write of the genes that should be corrected.
Line 333 and 335, there are mistakes when the authors cited the supplementary figures in text, the authors needs to carefully check it thorough the whole text
Line 343-348, Authors should follow the Instructions to Authors Guide, the full name of authors is needed.
The incited references in the text should be written in italic. The author should double check the format of the references (Line 367-479). The full name of magazine is required. The author needs to make the revision though all cited references.
Legend of figure 1 should be clearly state using letters instead of left or right.

Reviewer 3 ·

Basic reporting

Literature references, sufficient field background/context provided.

Experimental design

Research question well defined, relevant & meaningful.

Validity of the findings

Although the authors claimed comparative transcriptomic analysis was carried out, however, accessibility to raw data is not availiable.

Additional comments

The manuscript by Tingchun Li et al. titled “Comparative transcriptome analysis of differentially expressed genes related to the physiological changes of yellow-green leaf mutant of maize” isolated a new yellow-green leaf mutant. Subsequently, chlorophyll contents, chlorophyll fluorescence parameters, and photosynthesis characteristics were determined. In addition, comparative transcriptomic analysis was performed and 21 genes involved in chlorophyll metabolism, tricarboxylic acid cycle and sucrose to starch conversion expressed differentially between wild-type and the mutant. Based on the original and meaningful work, the manuscript is acceptable. However, the presentation should be improved since there were so many demerits as listed following:

1)Although the authors claimed comparative transcriptomic analysis was carried out, however, accessibility to raw data is not availiable. I suggest authors deposite their raw data to public database and provide the accession number in their manuscript to make their analysis repeatable.
2)Are biological repeats for both wild-type and mutant applied in RNA-Seq? Sampling is not described in details, thus I’m not convinced by their DGE analysis results.
3)At line 141, the title mentioned qRT-PCR, but qRT-PCR protocol was not described bellow.
4)Diffenretial genes were identified based merely on DGE anlysis and no experimental validation was performed. I suggest the authors quantify the expression of those genes by qRT-PCR.
5)The relationship between physiological changes and gene expression was not well discussed, thus the discussion section should be revised.

Reviewer 4 ·

Basic reporting

1.There are no accession number of Genbank for transcriptome data in paper, if the data will be uploaded?
2.It is necessary to make clear for the notes of figures, eg. A, B and C should be locatted on the upper left corner. And suggesting that Fig. 2 and Fig. 3 should be merged.

Experimental design

1.Chlorophyll is the main pigment in plant photosynthesis. And maize is an important food crop in the world, increasing the yield of maize is one of most important goals in breeding. Therefore, whether it can supplement the correlation analysis between the content of chlorophyll and yield? Whether the yield of yellow-green leaf is lower than green leaves? In order to achieve the purpose of increasing yield of maize, finding the yellow leaf genes and transfering the yellow leaf genes to green leaf by gene knockout or Crisper cas9 technology can be used in further study. It will be meaningful in molecular breeding for maize.
2.Specifically, the change of yellow green leaves occurred in which period of maize development? Could you show the specific spatiotemporal changes in phenotype of yellow-green leaf?
3.It is necessary to specify the period of leaf as the material. Why only choose one stage of leaf as the material? Suggesting that choosing more stages to analyze transcriptome, it will be better to find DEGs.
4.Why only choose two sets of datas for correlation analysis?

Validity of the findings

1.Many differentially expressed genes in chlorophyll synthesis pathways were found by transcriptomic analysis. The differentially expressed genes were verified by RT-PCR, but further verification by methods such as transgene or gene knockout or Crisper cas9 were still needed to ensure the reliability of the detected differentially expressed genes.
2.How to contact the physiological data and DEGs to explain the difference of yellow-green leaf and normal green leaf ?
3.How to regulate and control the formation of leaf colour between DEGs of many metabolic pathways, conjointly?
4.There are no integrated conclusion by combining the every passway to analyze the mechanism of the formation for yellow-green leaf.

Additional comments

This study analyzed the differentially expressed genes related to the physiological changes of yellow-green leaf mutant of maize by comparative transcriptome analysis. A number of genes in chlorophyll biosynthesis, photosynthesis, tricarboxylic acid cycle, and sucrose-to-starch conversion were found to be differentially expressed in the yellow-green leaf mutant and the normal green leaf inbred line. These findings provide potential explanations for observed morphological and physiological changes in the yellow-green leaf mutant. I have the following comments or suggestion for the author's reference.

---

## Round 0.2 · Minor Revisions

This is Julin Maloof, I have taken over as the Academic Editor for the manuscript. Thank you for addressing the reviewers' concerns. In looking over the manuscript I have one question. You describe that you normalized read counts via RPKM and then used EdgeR to adjust the read counts before going on to DEseq. My question is: what was your input to EdgeR? RPKM or the raw counts? It needs to be the raw counts. Please reanalyze if necessary and clarify either way. Also I think "chlorophyll contents" should be "chlorophyll content" throughout.

Reviewer 1 ·

Basic reporting

no comment

Experimental design

no comment

Validity of the findings

no comment

Additional comments

The manuscript is acceptable after revising.

·

Basic reporting

NO

Experimental design

NO

Validity of the findings

NO

Additional comments

The authors have made careful corrections to their manuscript, and all my advices have been accepted in the revised manuscript. I recommend it to be published.

Reviewer 4 ·

Basic reporting

The manuscript was well written and well organized.

Experimental design

The research question well defined, relevant and meaningful.

Validity of the findings

All underlying data have been provided. The findings are reliable.

---

## Round 0.3 · Minor Revisions

The section editor had the following comments and these need to be addressed before the paper is accepted:

1) I feel it is important that the authors make distinctions of whether the genes they are identifying are chromosomal- or plastid-borne; they really don’t address this at all, as traits can be tied to the plastid, or a nuclear-plastid interaction.

2) It is mentioned that GO terms were assigned; however, it is difficult to find the connections in the tables and supplemental files; this needs clarity.

3) There should be a ready available path to get to the 1122 genes discussed in the manuscript; either the raw set of sequence data, or the coordinate data from the maize reference genome. The only table that gets close to this is supplement table S2, but is is poorly constructed, as the ‘Novel’ sequences which are not available, and the Zm000xxxx genes would be aided by reference coordinates, and associated GO:12345 annotations.

4) Likewise, the KEGG data provided in the Figure 3 would benefit from a tabular form is supplemental data for reader validations, or better as an additional columns in table S2.

I would classify this manuscript as requiring moderate revision to bring more clarity to the topic. It does appear to fulfill the three replicate requirement desired from the 8 sequencing runs provided in the SRA archive. It does identify candidates, but in a fashion poorly for the reader to build upon.

---

## Round 0.4 · accepted · Accept

Thank you for providing the organized tables for GO and KEGG terms and the sequences of the novel genes.